Manuscript prepared for Ann. Geophys.
with version 1.3 of the LaTeX class copernicus.cls.
Date: 5 June 2020

# Magnetometer in-flight offset accuracy for the BepiColombo spacecraft

Daniel Schmid[1], Ferdinand Plaschke[1], Yasuhito Narita[1], Daniel Heyner[2], Johannes Z. D. Mieth[2], Brian J. Anderson[3], Martin Volwerk[1], Ayako Matsuoka[4], and Wolfgang Baumjohann[1]

[1]Space Research Institute, Austrian Academy of Sciences, Graz, Austria
[2]Institut für Geophysik und extraterrestrische Physik, Technische Universität Braunschweig, D-38106 Braunschweig, Germany
[3]The Johns Hopkins University Applied Physics Laboratory, Laurel, Maryland, USA
[4]Institute of Space and Astronautical Science, JAXA, Yoshinodai, Kanagawa, Japan

*Correspondence to:* Daniel Schmid, Space Research Institute, Austrian Academy of Sciences, Schmiedlstr. 6, 8042 Graz, Austria
(Daniel.Schmid@oeaw.ac.at)

**Abstract.** Recently the two-spacecraft mission BepiColombo launched to explore the plasma and magnetic field environment of Mercury. Both spacecraft, the Mercury Planetary Orbiter (MPO) and the Mercury Magnetospheric Orbiter (MMO, also referred to as Mio), are equipped with fluxgate magnetometers, which have proven to be well-suited to measure the magnetic field in space

with high precisions. Nevertheless, accurate magnetic field measurements require proper in-flight calibration. In particular the magnetometer offset, which relates relative fluxgate readings into an absolute value, needs to be determined with high accuracy. Usually, the offsets are evaluated from observations of Alfvénic fluctuations in the pristine solar wind, if those are available. An alternative offset determination method, which is based on the observation of highly compressional fluctuations

instead of incompressible Alfvénic fluctuations, is the so-called mirror mode technique. To evaluate the method performance in the Hermean environment, we analyze four years of MESSENGER (MErcury Surface, Space ENvironment, GEophysics and Ranging) magnetometer data, which are calibrated by the Alfvénic fluctuation method, and compare it with the accuracy and error of the offsets determined by the mirror mode method in different plasma environments around Mercury. We

show that the mirror mode method yields the same offset estimates and thereby confirms its applicability. Furthermore, we evaluate the spacecraft observation time within different regions necessary to obtain reliable offset estimates. Although the lowest percentage of strong compressional fluctuations are observed in the solar wind, this region is most suitable for an accurate offset determination with the mirror mode method. 132 hours of solar wind data are sufficient to determine the offset to

within $0.5\,\mathrm{nT}$, while thousands of hours are necessary to reach this accuracy in the magnetosheath

or within the magnetosphere. We conclude that in the solar wind the mirror mode method might be a good complementary approach to the Alfvénic fluctuation method to determine the (spin-axis) offset of the Mio magnetometer.

## 1 Introduction

In October 2018, BepiColombo, a two-spacecraft mission of the European Space Agency (ESA) and the Japan Aerospace Exploration Agency (JAXA), was launched to explore Mercury (Benkhoff et al., 2010). One of the spacecraft is the Mercury Planetary Orbiter (MPO), which is a 3-axis stabilized satellite (quasi nadir pointing) to study the surface and interior of the planet (e.g. Glassmeier et al., 2010). The other is Mio (or Mercury Magnetospheric Orbiter, MMO), a spin-stabilized

spacecraft (spin period of about $4\,\mathrm{s}$) to investigate the magnetic field environment of Mercury (e.g. Hayakawa et al., 2004; Baumjohann et al., 2006). During the $7.2\,\mathrm{year}$ cruise phase, both orbiters are transported by the Mercury Transfer Module (MTM) as a single composite spacecraft. In late 2025, the composite spacecraft will approach Mercury, where the MTM separates from the other two spacecraft, which are captured into a polar orbit around the planet. As soon as Mio reaches its

initial operational orbit of $590\,\mathrm{km}$ by $11640\,\mathrm{km}$ above the surface, also MPO separates and lowers its altitude to its $480\,\mathrm{km}$ by $1500\,\mathrm{km}$ orbit.

The BepiColombo Mercury Magnetometers (MERMAG) constitute a key experiment of the mission; MERMAG consists of the fluxgate magnetometers onboard both, MPO and Mio. The mag-

40 netometers will provide in-situ data for the characterization of the internal field origin as well as its dynamic interaction with the solar wind (see e.g. Wicht and Heyner, 2014, for a discussion). To achieve this goal, accurate magnetic field measurements are thus of crucial importance. Therefore, the components of a linear calibration matrix $\underline{\mathbf{M}}$ and an offset vector $\mathbf{O}$ need to be obtained, in order to convert raw instrument outputs $\mathbf{B}_{\mathrm{raw}}$ to fully calibrated magnetic field measurements (see e.g.

Kepko et al., 1996; Plaschke and Narita, 2016):

$$\mathbf{B} = \underline{\mathbf{M}} \cdot \mathbf{B}_{\mathrm{raw}} - \mathbf{O}. \tag{1}$$

Here, the matrix $\underline{\mathbf{M}}$ transforms $\mathbf{B}_{\mathrm{raw}}$ into a spacecraft-fixed orthogonal coordinate system. It comprises 9 parameters: 3 scaling (gain) values of the sensor and an orthogonalization matrix, which is defined by the 6 angles that yield the magnetometer sensor directions with respect to the spacecraft

reference frame (see e.g. Plaschke and Narita, 2016). The 3D offset vector $\mathbf{O}$, on the other hand, reflects the magnetometer outputs in vanishing ambient fields. These can be attributed to the instrument and also to the field generated by the spacecraft at the position of the magnetometer sensor. Frequent in-flight calibration of these offsets is necessary, as they are known to change over time.

To calibrate the magnetometer, all 12 parameters need to be accurately determined. For spinning

spacecraft (i.e. Mio) 8 of the calibration parameters can be determined directly by minimizing pe-

riodic signatures in the de-spun magnetic field signal at the spin frequency and/or at the second harmonic (Kepko et al., 1996). The remaining 4 parameters i.e. the absolute gain in the spin-plane and along spin-axis, the rotation angles of the sensor around the spin-axis, and the spin-axis offset need to be determined differently. It should be noted that the gains and rotation angle become im-

60 portant at strong fields. In the solar wind and in the Hermean environment they may play a minor role in the magnetic field measurements accuracy in comparison to the offsets.

The following methods, which can also be applied to non-spinning spacecraft (i.e. MPO), are well established for the offset determination:

(1) Cross-calibration of the magnetometer offset with independent magnetic field measurements from other instruments. The Magnetospheric Multiscale mission (MMS, Burch et al., 2016) use independent measurements from the Electron Drift Instruments (EDI, Torbert et al., 2016) to cross-calibrate the spin-axis offset of the magnetometers (Nakamura et al., 2014; Plaschke et al., 2014)).

(2) The offset may also be determined in a characteristic region where the magnetic field is known.

Goetz et al. (2016) used diamagnetic cavities to determine the magnetometer offsets of the Rosetta spacecraft mission (Glassmeier et al., 2007).

(3) Spacecraft rotations about the spacecraft principal axes from time to time are also a well-established method to obtain magnetometer offsets. This has been done routinely with many missions, most recently including MAVEN (Connerney et al., 2015) at Mars and Parker Solar Probe

(Bale et al., 2016).

(4) A common way to calibrate the magnetometer offset in-flight, is to use (a) (nearly-)incompressible or (b) compressible fluctuations of the magnetic field:

4(a) A well-established method is to minimize the variance of the total magnetic field during the passage of (nearly-)incompressible (Alfvénic) variations in the (pristine) solar wind (Belcher, 1973;

Hedgecock, 1975). Pure Alfvénic fluctuations are strictly incompressible and circularly polarized. They are characterized by changes in the magnetic field components while the magnitude of the field stays constant. Particularly in inhomogeneous media such simple classifications are found to be impossible (see Tsurutani et al., 2018, for a review). However, in the solar wind the fluctuations of the magnetic field strength (compressible part) are weak compared to the strong fluctuations of the mag-

netic field vector direction (Khabibrakhmanov and Summers, 1997). By minimizing the changes of the observed total magnetic field of such fluctuations it is therefore possible to adjust the magnetometer offsets with high precision. In Figure 1(a) a schematic illustration of (nearly-)incompressible Alfénic fluctuations is shown. The changes of the total magnetic field, $\delta|B|$, observed by a virtual spacecraft crossing the fluctuations are minimal (shown in the bottom).

4(b) Recently Plaschke and Narita (2016) introduced an alternative magnetometer offset calibration technique on the basis of the observation of compressible fluctuation, the so-called mirror mode method, which does not require pristine solar wind measurements. The idea is that for strongly com-

pressible fluctuating fields (e.g. mirror modes) the maximum variance direction of the magnetic field should be nearly parallel to the mean (background) magnetic field (Tsurutani et al., 2011). Mirror

mode structures are compressional, non-propagating structures which have been observed in various space plasma environments like in the solar wind (e.g. Winterhalter et al., 1994), planetary magnetosheaths (e.g. Tsurutani et al., 1982) and even near the magnetic pileup boundaries of comets (e.g. Glassmeier et al., 1993). They are typically characterized by the anticorrelation between the magnetic field strength and density fluctuations, with little or no change in the magnetic field direction

across the structure (see e.g. Winterhalter et al., 1994). Figure 1(b) shows a sketch of such a mirror mode structure. A virtual spacecraft crossing the structure observes perturbations along the mean magnetic field direction, $\delta B_{||}$. Any differences between the maximum variance direction (evaluated from e.g. minimum variance analysis (MVA), Sonnerup and Scheible, 1998) of the magnetic field and the mean (background) magnetic field direction can thus be used for the magnetometer offset

determination. Note that although this method is called mirror mode method, observations of highly compressional waves other than mirror mode structures with strong $\delta|B|/|B|$ ratios without $\mathbf{B}$ direction changes are sufficient for their application. An advantage of this method is that compressible waves are ubiquitous in the magnetosphere and magnetosheath. Therefore this method can also be applied to calibrate the magnetometers of spacecraft which remain within the magnetosphere (like

e.g. MPO).

In this paper we test the applicability of the mirror mode method in different regions (plasma environments) around Mercury, based on 4-years of MESSENGER (MErcury Surface, Space ENvironment, GEophysics and Ranging, Solomon et al., 2007) fluxgate magnetometer data (FGM, An-

115 derson et al., 2007), which were calibrated using time intervals of (nearly-)incompressible Alfvénic fluctuations in the solar wind. For reasons of simplicity (and because for Mio only the spin-axis offset needs to be evaluated) we apply the 1-D mirror mode method (see Plaschke and Narita, 2016) and assume that two offset components are already accurately determined. We test to which degree the 1-D mirror mode method yields vanishing offsets as expected when using calibrated data as in-

120 put. We also address the question of how much time a spacecraft needs to spend in each individual plasma region until the magnetometer offset can be determined to a specific accuracy from intervals containing compressional magnetic field fluctuations.

The results obtained in this paper enable us to assess whether the mirror mode method would be a useful tool to accurately determine the offset of BepiColombo's magnetometer.

## 2  Data and Methodology

We use orbital magnetic field data from the MESSENGER fluxgate magnetometer (FGM, Anderson et al., 2007) between March 2011 and April 2015. The polar orbit of MESSENGER was highly

elliptical; the initial altitude ranged between $200\,\mathrm{km}$ and $15000\,\mathrm{km}$ form Mercury's surface. With
an initial orbital period of 12 hours, MESSENGER crossed the magnetopause and bow shock four
times within 24 hours. We use the 1-Hz calibrated magnetic field data in spacecraft coordinates,
where the $Y$-axis is nominally in the anti-sunward direction (radially away from the Sun), the $Z$-axis
points towards the payload adapter ring at the bottom of the spacecraft, and the $X$-axis completes
the right handed coordinated system. If not noted otherwise, we use the magnetic field components
$\{B_{\mathrm{x}}, B_{\mathrm{y}}, B_{\mathrm{z}}\}$ throughout this paper in these coordinates. MESSENGER was a three-axis-stabilized
spacecraft, and its magnetometer offsets were routinely corrected using time intervals of Alfvénic
fluctuations in the solar wind. We therefore use the MESSENGER magnetometer data as a calibra-
tion standard in our magnetometer offset study. To perform a test of the the mirror mode method
against the MESSENGER magnetometer data, we determine the 1D offset along the $Z$-axis, $O_{\mathrm{z}}$, in
the same way as introduced by Plaschke and Narita (2016):

Within strongly compressible mirror mode structures, the magnetic field variation maximum vari-
ance direction, $\boldsymbol{\ell} = [l_{\mathrm{x}}, l_{\mathrm{y}}, l_{\mathrm{z}}]$ obtained from a principal component analysis (minimum variance
analysis, MVA, Sonnerup and Scheible, 1998) should be reasonably aligned with the mean field
direction, $\mathbf{B} = [B_{\mathrm{x}}, B_{\mathrm{y}}, B_{\mathrm{z}}]$. Under assumption of alignment between the maximum variance di-
rection and the mean field direction Plaschke and Narita (2016) showed that the offset $O_{\mathrm{z}}$ can be
derived by:

$$O_{\mathrm{z}} = B_{\mathrm{xy}}(\tan\theta_{\mathrm{B}} - \tan\theta_{\ell}), \tag{2}$$

where $\theta_{\mathrm{B}} = \arctan(B_{\mathrm{z}}/B_{\mathrm{xy}})$ is the elevation angle of magnetic field to the $XY$-plane and
$\theta_{\ell} = \arctan(\ell_{\mathrm{z}}/\ell_{\mathrm{xy}})$ the elevation angle of the maximum variance direction to that plane. $B_{\mathrm{xy}} = \sqrt{B_{\mathrm{x}}^2 + B_{\mathrm{y}}^2}$ and $\ell_{\mathrm{xy}} = \sqrt{\ell_{\mathrm{x}}^2 + \ell_{\mathrm{y}}^2}$ are the magnetic field and maximum variance within the $XY$-
plane.

Since a single offset estimate alone might not be very accurate, a statistically significant offset $O_{\mathrm{zf}}$
should be determined by finding the maximum of the probability density function $P$ computed by
the kernel density estimator (KDE) method with Gaussian kernel from a sample of individual offset
estimations $O_{\mathrm{z}}$:

$$P(O_{\mathrm{z}}^*) = \frac{1}{\sqrt{2\pi}Nh} \sum_{n=1}^{N} \exp\left[-\frac{1}{2}\left(\frac{O_{\mathrm{z}}^* - O_{\mathrm{z,n}}}{h}\right)^2\right]. \tag{3}$$

Here $N$ denotes the number of individual offset estimates, $O_{\mathrm{z,n}}$, $h$ is a smoothing parameter which
denotes the bandwidth of the KDE, and $O_{\mathrm{zf}}$ is the offset value at which the probability $P$ is at max-
imum, $\max\left(P(O_{\mathrm{z}}^*)\right) = P(O_{\mathrm{zf}})$. We use the method introduced by Silverman (1986), to determine
an optimal bandwidth $h = c_1 \cdot \sigma(O_{z,n})N^{c_2}$, with $c_1 = 1.06$ and $c_2 = -1/5$. The symbol $\sigma$ denotes
the standard deviation of the offset-value distribution.

## 3 Data Analysis and Results

A basic condition for the mirror mode method, is the availability of compressional magnetic field fluctuations. As a first step we use the magnetometer data to estimate the occurrence rate of compressional fluctuations in the different plasma environments around Mercury. Then we compare the offset determination by the mirror mode method with the Alfvén wave method in terms of accuracy. Finally, we evaluate the number of offset samples that are sufficient for in-flight calibration with the mirror mode method, given a minimum required accuracy. Therewith, we subsequently determine how many hours the spacecraft (i.e. Mio and/or MPO) need to spend in different plasma environments to obtain reliable offset values.

### 3.1 Occurrence rate of compressional fluctuations

Mercury's plasma environment is highly dynamic and home to a plethora of wave modes and fluctuations (Russell, 1989; Boardsen et al., 2009, 2012; Sundberg et al., 2015). We separate the MESSENGER orbit segments into solar wind, magnetosheath and magnetospheric parts, based on an extended boundary data set (Winslow et al., 2013, and personal communication with R. Winslow and B. J. Anderson), in order to distinguish occurrence rates of (compressional) fluctuations and offset estimate accuracies by region. We characterize the observed fluctuations in the following way: The magnetic field data are divided into overlapping 30-s intervals shifted by 15-s. Within each sub-interval the magnetic field measurements are transformed into a mean-field-aligned (MFA) coordinate system, where the parallel component, $\hat{\mathbf{b}}_{||} = \mathbf{B}_0/|\mathbf{B}_0|$, is given by the average magnetic field within the 30-s interval, $\mathbf{B}_0 = [B_{\mathrm{x},0}, B_{\mathrm{y},0}, B_{\mathrm{z},0}]$, and the perpendicular components in this coordinate system are chosen to be $\hat{\mathbf{b}}_{\perp,1} = [0, -B_{\mathrm{z},0}, B_{\mathrm{y},0}]/|[0, -B_{\mathrm{z},0}, B_{\mathrm{y},0}]|$ and $\hat{\mathbf{b}}_{\perp,2} = \hat{\mathbf{b}}_{\perp,1} \times \hat{\mathbf{b}}_{||}$. Then the maximum variance direction, $\ell$, of the transverse (perpendicular) magnetic field components is evaluated by a 2D minimum variance analysis (MVA, Sonnerup and Scheible, 1998). The difference between the maximum and minimum values of the magnetic field in this maximum variance direction yields the measure for the transverse fluctuations $\delta B_{\perp,\ell} = B_{\perp,\ell}^{\max} - B_{\perp,\ell}^{\min}$. The compressional fluctuations, on the other hand, are essentially represented by fluctuations in the magnetic field magnitude $\delta|B|$, given by the difference between the maximum and minimum value of the magnetic field magnitude within the 30-s interval. Subsequently, to determine whether the compressional or the transverse part is dominating, we further define the compressibility index $Q^\pm$ as

$$Q^\pm = \log_{10}\left(\frac{\delta|B|}{\delta B_{\perp,\ell}}\right), \tag{4}$$

which is positive/negative in case of dominating compressional/transverse fluctuations.

Left panel of Figure 2 displays the occurrence rate (normalized to unity when integrated over the whole domain) of fluctuations of the magnetic field magnitude (compressible sense) relative to the mean field in various plasma regions: in the solar wind (in red), in the magnetosheath (in

green), and in the Hermean magnetosphere (in blue) for the MESSENGER magnetometer data for 4 years. Most of the large-amplitude fluctuations are observed in the magnetosheath. Such a result is not surprising, since the magnetosheath is characterized generally by a highly turbulent plasma with enhanced magnetic field variations. The right panel shows the normalized occurrence rate of the compressibility index $Q^\pm$ of selected intervals with large amplitude fluctuations ($\delta|B|/\overline{B} > 0.3$). The shifts of the maximum occurrence rates to negative $Q^\pm$ values indicate that statistically transverse fluctuations were dominating. However, there is also a significant number of time intervals with compressional fluctuations observed. The integrated occurrence rates are shown in Table 1. The first column of Table 1 shows the percentage of the sub-intervals where the magnetic field magnitude fluctuations $\delta|B|/\overline{B}$ were larger than 0.3. The percentage in the second column shows how many of the intervals with large-amplitude fluctuations ($\delta|B|/\overline{B} > 0.3$) are dominated by strongly compressional fluctuations, where $\delta|B|/\delta B_{\perp,\ell} > 2$ ($Q^\pm > 0.3$). The last column reflects the percentage of intervals that include these strongly compressional fluctuations out of all intervals of MESSENGER observations in the respective regions.

About $25\%$ of the time enhanced magnetic field fluctuations ($\delta B/\overline{B} > 0.3$) are observed in the magnetosheath and $10.6\%$ of these fluctuations had a compressional nature. That means that from the total observation time in the magnetosheath, $2.6\%$ of the time MESSENGER observed strongly compressional fluctuations. A smaller fraction of the MESSENGER data set represent compressible dominated time intervals of the solar wind ($0.4\%$) and the Hermean magnetosphere ($1.7\%$). This suggests that the MESSENGER magnetosheath data may be best suited for the offset calibration with the mirror mode method.

### 3.2 Test of the mirror mode method

Using Equation 2 we determine the offset $O_\mathrm{z}$ within each 30-sec time interval. To ensure reliable offset estimates, the same requirements introduced by Plaschke and Narita (2016) have to be fulfilled within each window:

- $\frac{\delta B_\mathrm{xy}}{\overline{B}_\mathrm{xy}} = \frac{B_\mathrm{xy}^\mathrm{max} - B_\mathrm{xy}^\mathrm{min}}{B_\mathrm{xy}^\mathrm{mean}} > 0.3$, since mirror modes are characterized by large magnetic field fluctuations (see Price et al., 1986; Schmid et al., 2014). $B_\mathrm{xy}^\mathrm{max}$, $B_\mathrm{xy}^\mathrm{min}$ and $B_\mathrm{xy}^\mathrm{mean}$ are the maximum, minimum and average magnetic field values in the spacecraft $XY$-plane, respectively. Note that we only consider the $XY$-plane components, because the $Z$-component are subject to an a-priori unknown offset $O_\mathrm{z}$.

- $\phi < 20°$, where $\phi$ is the angle between the maximum variance, $\ell$, and magnetic field, $\mathbf{B}$, directions in the $XY$-plane. Note that only the $XY$-plane components is used. Plaschke and Narita (2016) derived this requirement and threshold from Lucek et al. (1999), who identified mirror modes using the angle between maximum variance and magnetic field direction.

- $|\theta_\mathrm{B}| < 30°$ and $|\theta_\ell| < 30°$, so that both, maximum variance and magnetic field directions

point closer to the $XY$-plane.

Out of 4-years of MESSENGER data these requirements are met for $3.0\,\%$, $2.1\,\%$ and $0.4\,\%$ of the intervals within the magnetosphere, magnetosheath and the solar wind, respectively (see also first column of Table 2). It should be noted that these numbers do not have to match the numbers shown in Table 1, since the criteria are different. The numbers in Table 1 reflect the occurrences of strongly compressional fluctuations and here they reflect the occurrences of intervals with mirror mode like characteristics. Interestingly, however, the numbers in the solar wind and the magnetosheath are similar but differ considerably in the magnetosphere.

Using Equation 3 the probability density function $P$ is computed for each region from all determined best-estimate offset $O_z$ in that region. Under the assumption that the data are perfectly calibrated and the mirror mode method works accurately, the offset $O_{zf}$ should vanish and $P$ should be highly symmetric with the peak around $0\,\mathrm{nT}$.

Figure 3 shows the probability density functions $P$ based on the magnetosphere (red), magnetosheath (green) and solar wind (blue) offset estimates. Indeed, the best estimate of the $Z$-component offset $O_{zf}$ is around zero for all three regions. It is noteworthy that the standard deviation of individual offset estimates, $O_{z,n}$, is smallest in the solar wind. This is actually not surprising, since the intervals in the solar wind where the above requirements are fulfilled, are well-suited for this calibration method due to their clear compressional signature in the rather low background magnetic field. In Table 2 the total number of intervals ($O_z$ estimates) and their best-estimate offset $O_{zf}$ are given. Further, the arithmetic mean of the individual offsets in each interval $O_{z,n}$ with their standard deviations and error are summarized.

The average $\langle O_{z,n} \rangle$ is found to be quite close to the final offset estimate $O_{zf}$. Note that, although the standard deviation of the individual offsets $O_{z,n}$ might be large, a larger number of samples or events helps lower the value of the standard deviation of the mean offset $\langle O_{z,n} \rangle$ (standard error in Table 2).

### 3.3 Effect of sample size on offset accuracy

In the following we check how the best-estimate offset $O_{zf}$ is affected by the number of $O_{z,n}$ estimates. Therefore, from each region, 1 to 20000 offset estimates ($N = \{x \cdot 10^y \mid x \in [1,9], y \in [0,4]\}$, $\max(N) = 20000$) are randomly picked and their best-estimate offset $O_{zf}$ determined. For each $N$, this sequence is repeated 1000 times (bootstrapping). The standard deviation of the 1000 $O_{zf}$ ($\sigma(O_{zf})$) subsequently reflects the uncertainty of the determined $O_{zf}$. Here we use the 2-$\sigma$ interval, to evaluate the uncertainty of the best-estimate offset $O_{zf}$ with $95\,\%$ confidence. The standard error $\epsilon$ of the 2-$\sigma$ uncertainty of the best-estimate offset, 2-$\sigma(O_{zf})$, from the bootstrapping is given by $\epsilon \simeq$ 2-$\sigma(O_{zf})/\sqrt{2(M-1)}$ (see e.g. Squires, 2001). Here $M = 1000$ yielding a relative standard error $\epsilon = 4.3\,\%$ of the evaluated 2-$\sigma(O_{zf})$. Figure 4 displays 2-$\sigma(O_{zf})$ as a function of the number $N$ of

the offset estimates $O_{\mathrm{z,n}}$ used.

The dashed lines in Figure 4 mark offset accuracies at $0.5\,\mathrm{nT}$ and $1.0\,\mathrm{nT}$. It is visible that the offset accuracy increases with the number of samples following a power law. However, below $\sim$ $0.5\,\mathrm{nT}$ considerably more samples are needed to improve the offset accuracy, which could indicate the lower limit of the offset accuracy of the MESSENGER magnetometer determined by the Alfvénic method. As can been seen in Figure 4, this divergence is only observed in the solar wind, because it is the only region where a significant number of $O_{\mathrm{zf}} < 0.5\,\mathrm{nT}$ are obtained. The solid lines in Figure 4 depict the regression lines of a linear least squares fit of the offset accuracies above $0.5\,\mathrm{nT}$ and clearly shows the power law behavior between these offset accuracies and the number of samples. Table 3 shows the fitting parameters of the regression lines of the power laws,

$$\log_{10}[2\sigma(O_{\mathrm{zf}})] = \log_{10}[a] + k \cdot \log_{10}[N]. \tag{5}$$

Here $a$ denotes the 2-$\sigma$ confidence of the best-estimate $O_{\mathrm{zf}}$ determined from only one offset $O_{\mathrm{z}}$ and $k$ represents the spectral index of the power law.

From the least squares approximation we calculate the minimum number $N$ of $O_{\mathrm{z,n}}$ estimates to reach an offset accuracy of $0.5\,\mathrm{nT}$ and $1.0\,\mathrm{nT}$. These numbers are multiplied by $30\,\mathrm{s}$, the sliding window time interval, yielding the time ranges from which reliable $O_{\mathrm{z,n}}$ estimates are obtained (i.e. where the criteria from above are satisfied). These time ranges multiplied with the probability to observe these time intervals (see first column in Table 2), finally give the time lengths that a spacecraft needs to spend in the solar wind, magnetosheath or magnetosphere in order to determine the offset with an accuracy better than $0.5\,\mathrm{nT}$ or $1.0\,\mathrm{nT}$. Table 4 shows the minimum number $N$ of $O_{\mathrm{z,n}}$ estimates with their corresponding time ranges and the necessary spacecraft observation time, required to reach these accuracies.

While 132 hours of solar wind data are sufficient to determine the offset at an accuracy of $0.5\,\mathrm{nT}$, 4241 hours are needed in the magnetosphere and more than 6130 hours in the magnetosheath. This is an interesting result, since the magnetosheath is the region with the highest probability to observe large-amplitude compressional fluctuations and the solar wind with the lowest (see Table 1). A possible explanation might be that the solar wind data are observations in the shock-upstream region in which the shock-reflected backstreaming ions excite large-amplitude Alfvén waves that either pitch the plasma (through ponderomotive force) or develop into magnetosonic-type waves. However, the results suggests that the fluctuations in the solar wind, although observed less often, are better suited to determine the offset with the mirror mode method.

## 4   Discussion and Conclusions

We find that the offset determination method proposed by Plaschke and Narita (2016) is well applicable to the data from the Hermean environment. The results reveal that the solar wind is the most suitable region to accurately determine the offset by the mirror mode method, although the lowest

percentage of highly compressible fluctuations are observed there (see Table 1). As can been seen in Table 4, offset determination with uncertainties better than $0.5\,\mathrm{nT}$ can be achieved with less than 132 hours of solar wind data. However, it is important to note that during this time the instrument offsets need to stay constant to within $0.5\,\mathrm{nT}$, otherwise the intrinsic offset drifts over time would limit the achievable precision, independently from the the amount of solar wind data. Specifically in the magnetosheath and within the magnetosphere this might be a more important limiting factor, since several thousands of hours of data in those regions are needed to ensure that the offset uncertainty is less than $0.5\,\mathrm{nT}$ (see Table 4). Figure 4 shows that the offset accuracies down to $0.5\,\mathrm{nT}$ diminish with the number of necessary $O_z$ estimates and follows a power law. However, below $0.5\,\mathrm{nT}$ the power law correlation flattens (spectral index becomes smaller), which might indicate that the lower limit of the offset accuracy of the calibrated magnetometer data itself has been reached. To ensure e.g. the offset uncertainty to be of the order of $1.0\,\mathrm{nT}$, 60 hours of magnetic field data in the solar wind would be sufficient, but hundreds of hours of magnetosheath or magnetosphere data are would be needed.

In the following we consider possible implications of these results on the BepiColombo mission:

– **Mio (Mercury Magnetospheric Orbiter, MMO):** Since Mio's orbit allows the spacecraft encounter the solar wind (particularly near Mercury's perihelion), the mirror mode method would serve as a reasonable complementary approach to the Alfvénic fluctuation method. Furthermore, Mio is a spin-stabilized spacecraft (with a period of about $4\,\mathrm{s}$) and thus two of the three offsets in the de-spun plane can be directly determined by minimization of spin-tone in the data. Since the two spin-plane offsets can be determined very accurate (Kepko et al., 1996), the mirror mode method presented in this study can thus be directly applied to determine the remaining offset (spin-axis offset) very precisely (see, Plaschke and Narita, 2016). However, the mirror mode method assumes the time independence of the offset properties during the fluctuation measurements. Offsets drifts over time, e.g. due to temperature changes over orbital periods, will be the limiting factors for the achievable accuracies.

– **MPO (Mercury Planetary Orbiter):** The MPO spacecraft, on the other hand, is 3-axis stabilized, which should diminish the offset determination accuracy due to the two additional degrees of freedom and the 1D mirror mode method presented in this study cannot be applied directly. Based on the principles of the 1D mirror method, Plaschke et al. (2017) extended the method to 3 dimensions and introduced the so-called 3D mirror mode method. This method has successfully been tested to 3-axis stabilized spacecraft magnetic field measurements and is able to determine the 3D offset vectors directly from observations of highly compressional magnetic field fluctuations. Hence, the 3D mirror mode method would be more appropriate to determine the MPO magnetometer offsets. However, the applicability of the 3-D mirror mode

method needs to be evaluated in detail and is beyond the scope of this paper. Nevertheless, from the 1D mirror mode method preformed in this work, it is possible to get a first idea of how many hours of magnetic field observations are needed in order to determine the offset of one component if the other two are well determined. The orbit of MPO will remain in the Hermean magnetosphere for most of the time (except for cusp crossings or during events of high dynamic pressure in the solar wind). Based on the results obtained from MESSENGER, 780 hours of observations are needed in order to determine the offset of one component with an accuracy better than $1.0\,\mathrm{nT}$ in this region. Note that during this long observation period the magnetometer offsets need to stay constant within $1.0\,\mathrm{nT}$, as mentioned above.

From the results obtained in this work we conclude that in the solar wind the mirror mode method should be a good complementary approach to the Alfvénic fluctuation method to determine the spin-axis offset of the Mio magnetometer. In case of MPO we cannot evaluate from this study whether the mirror mode method is sufficient to obtain reliable offset estimates. However, from the results of the 1D mirror mode method, we find that considerably more data are needed to reach the same offset accuracy, since the orbit nominally remains within Mercury's magnetosphere. In a future work the 3D mirror mode method should be applied on the MESSENGER data to asses whether the mirror mode method might also be a valuable tool to obtain reliable offset estimates also for the magnetometers of MPO-MAG.

*Data availability* The MESSENGER magnetic field (MAG) data are obtained from the NASA Planetary Data System (PDS). All data are open access and can be retrieved on the PDS website (https://pds-ppi.igpp.ucla.edu, NASA Planetary Date System, 2015).

*Author contributions* DS initiated this study, collected the data and did the analysis. FP wrote the initial version of the Mirror Mode Method program and helped interpreting the results. YN contributed Figure 1 and helped evaluating the manuscript and interpreting the results. WB, AM, DH and JM also assisted in evaluating the manuscript. As the PI of the MESSENGER magnetometer, BA guaranteed the quality and usability of the data.

*Competing interests* The authors declare that they have no conflict of interest.

*Financial support* This work is financially supported by the Austrian Research Promotion Agency (FFG) ASAP MERMAG-4 under contract 865967. The German Ministerium für Wirtschaft und Energie and the German Zentrum für Luft- und Raumfahrt supported the work of D. Heyner under contract 50 QW 1101 and the work of J. Z. D. Mieth under contract 50 OC 1403.

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

**Tables**

**Table 1.** The first column gives the percentage of the sub-intervals where the magnetic field magnitude fluctuations $\delta|B|/\overline{B}$ is larger than 0.3 in each respective region. The percentage in the second column shows how many intervals of the sub-intervals with $\delta|B|/\overline{B} > 0.3$ are dominated by compressional fluctuations ($Q^{\pm} > 0.3$). The last column reflects the percentage of compressional fluctuations MESSENGER observes in total in each region.

| | $\delta|B|/\overline{B} > 0.3$ | | compressible |
| Region | total | $Q^{\pm} > 0.3$ | fluctuations in total |
|---|---|---|---|
| Solar wind | 7.9 % | 5.4 % | 0.4 % |
| Magnetosheath | 24.6 % | 10.6 % | 2.6 % |
| Magnetosphere | 8.0 % | 21.6 % | 1.7 % |

**Table 2.** Number of samples (time intervals) used in the statistical study for the offset determination and various offset estimates in units of nT (best-estimate, mean, standard deviation, and standard error).

| Region | Number of samples (% of total) | best-estimate $O_{zf}$ [nT] | mean $\langle O_{z,n} \rangle$ [nT] | standard deviation $\sigma(O_{z,n})$ [nT] | standard error $\sigma(O_{z,n})/\sqrt{N}$ [nT] |
|---|---|---|---|---|---|
| Solar wind | 21200 (0.4 %) | $-0.04$ | $-0.10$ | 6.5 | 0.04 |
| Magnetosheath | 29289 (2.1 %) | $-0.17$ | $-0.34$ | 14.0 | 0.08 |
| Magnetosphere | 36652 (3.0 %) | $-0.01$ | $-1.01$ | 12.4 | 0.09 |

**Table 3.** Fitting parameters of Equation 5 with 95 % confidence intervals, exhibited from linear least squares fit of the offset accuracies above $0.5\,\mathrm{nT}$ in Figure 4. $a$ is the 2-$\sigma$ confidence of the best-estimate $O_{zf}$ determined from only one offset $O_z$ and $k$ is the spectral index of the power law.

| Region | $a$ [nT] | $k$ |
|---|---|---|
| Solar wind | $18.6 \pm 1.1$ | $-0.87 \pm 0.03$ |
| Magnetosheath | $34.8 \pm 1.0$ | $-0.44 \pm 0.01$ |
| Magnetosphere | $25.9 \pm 1.0$ | $-0.41 \pm 0.01$ |

**Table 4.** Minimum number of $O_{z,n}$ estimates (first row) and corresponding time ranges (second row) required to determine the offset with an accuracy of $0.5\,\mathrm{nT}$ or $1.0\,\mathrm{nT}$ in the solar wind, magnetosheath or magnetosphere. The necessary observation time of the spacecraft (third row) in each region.

| | Solar Wind | | Magnetosheath | | Magnetosphere | |
| Offset accuracy | 0.5 nT | 1.0 nT | 0.5 nT | 1.0 nT | 0.5 nT | 1.0 nT |
|---|---|---|---|---|---|---|
| minimum number of samples | 63 | 29 | 15325 | 3173 | 15422 | 2837 |
| time to reach accuracy | 31 min | 14 min | 128 h | 26 h | 128 h | 24 h |
| necessary S/C observation time | 132 h | 60 h | 6130 h | 1269 h | 4241 h | 780 h |

**Figure Captions**

**Fig. 1.** Schematic illustration of (a) Alfvénic fluctuations and (b) mirror mode fluctuations. The bottom panels show the in-situ (a) magnetic magnitude fluctuations, $\delta|B|$ and (b) mean (background) magnetic field fluctuations $\delta|B_{||}|$ of a virtual spacecraft crossing the respective region.

**Fig. 2.** Left: Normalized occurrence rate of the total magnetic field fluctuations $\delta|B|/\overline{B}$ in the solar wind (blue), magnetosheath (green) and magnetophere (red) during 4 years of MESSENGER observations. Right: Normalized occurrence rate of the compressibility index $Q^{\pm}$ for magnetic fluctuations with $\delta|B|/\overline{B} > 0.3$. Positive or negative $Q^{\pm}$ values indicate whether the compressional or the transverse part of the magnetic fluctuations is dominating and $Q^{\pm} > 0.3$ values are considered as strong compressional fluctuations.

**Fig. 3.** Probability density function $P$, estimated with the KDE method from Equation 3 from 4-years of MESSENGER observations.

**Fig. 4.** The relationship between 2-$\sigma(O_{\mathrm{zf}})$ as a function of the number $N$ of the offset estimates $O_{\mathrm{z,n}}$. 2-$\sigma(O_{\mathrm{zf}})$ is the uncertainty of the determined best-estimate offset $O_{\mathrm{zf}}$ with 95 % confidence. The solid lines represent the linear least squares fits of the 2-$\sigma$ confidence offsets above $0.5\,\mathrm{nT}$.

**Figures**

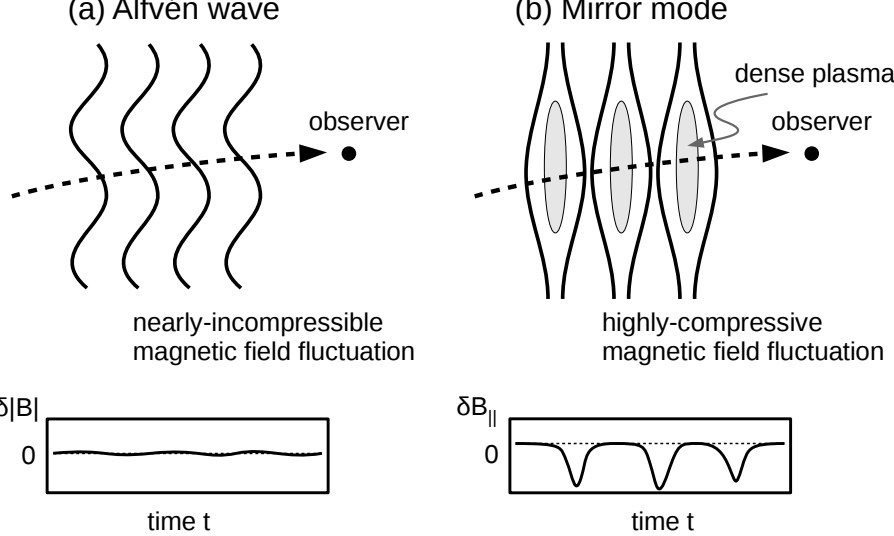

**Fig. 1.** Schematic illustration of (a) Alfvénic fluctuations and (b) mirror mode fluctuations. The bottom panels show the in-situ (a) magnetic magnitude fluctuations, $\delta|B|$ and (b) mean (background) magnetic field fluctuations $\delta|B_{||}|$ of a virtual spacecraft crossing the respective region.

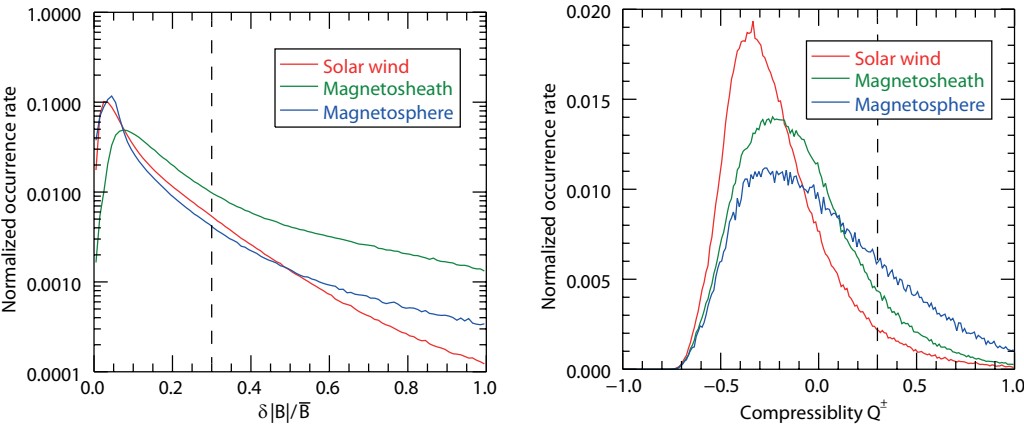

**Fig. 2.** Left: Normalized occurrence rate of the total magnetic field fluctuations $\delta|B|/\overline{B}$ in the solar wind (blue), magnetosheath (green) and magnetophere (red) during 4 years of MESSENGER observations. Right: Normalized occurrence rate of the compressibility index $Q^{\pm}$ for magnetic fluctuations with $\delta|B|/\overline{B} > 0.3$. Positive or negative $Q^{\pm}$ values indicate whether the compressional or the transverse part of the magnetic fluctuations is dominating and $Q^{\pm} > 0.3$ values are considered as strong compressional fluctuations.

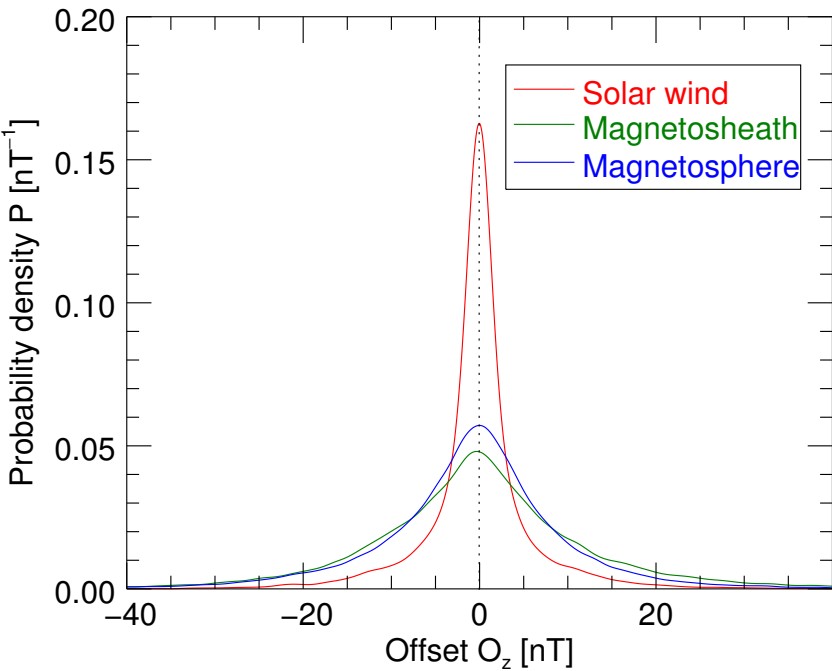

**Fig. 3.** Probability density function $P$, estimated with the KDE method from Equation 3 from 4-years of MESSENGER observations.

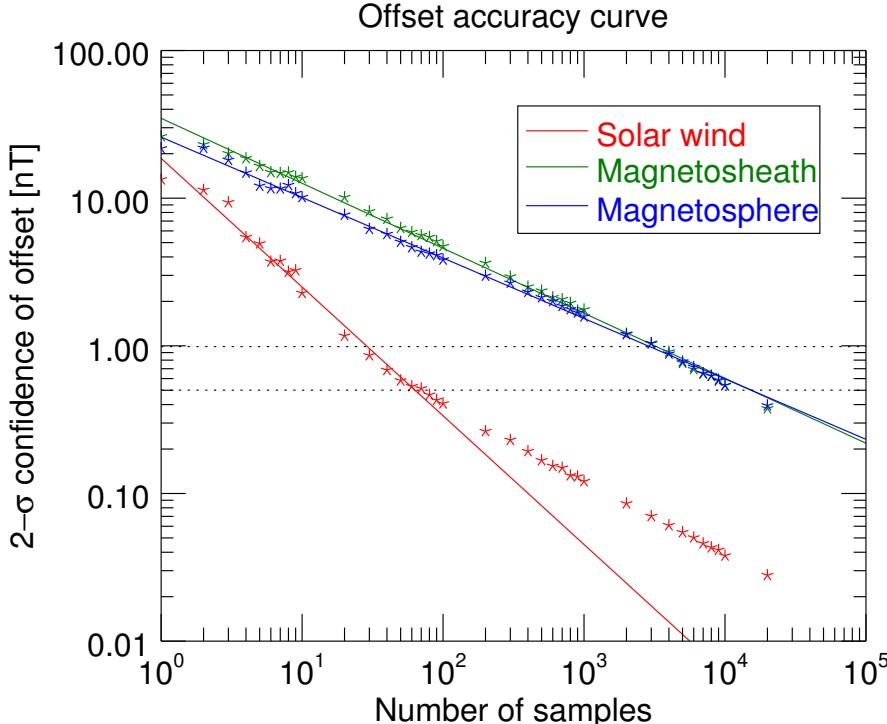

**Fig. 4.** The relationship between $2\text{-}\sigma(O_{zf})$ as a function of the number $N$ of the offset estimates $O_{z,n}$. $2\text{-}\sigma(O_{zf})$ is the uncertainty of the determined best-estimate offset $O_{zf}$ with $95\,\%$ confidence. The solid lines represent the linear least squares fits of the $2\text{-}\sigma$ confidence offsets above $0.5\,\text{nT}$.