# Peer review of "Magnetometer in-flight offset accuracy for the BepiColombo spacecraft"

_Annales Geophysicae, 2020_

## Referee Comment (RC1) · Anonymous Referee #1 · 26 Mar 2020

The submitted manuscript presents an approach to calibrating offsets on the magnetometers onboard BepiColombo's Mercury Planetary Orbiter and Mercury Magnetospheric Orbiter spacecraft. This calibration analysis includes the use of mirror mode wave observations as a method to determine the spin-axis offset. Mio would be able to utilize this approach as a complementary method to an analysis of Alfvenic fluctuations in the pristine solar wind. MPO, on the other hand, will not measure the solar wind and therefore, observations within Mercury's magnetosphere must be used to calibrate measurements. The manuscript presents an analysis of the compressional fluctuations in Mercury's space environment by analyzing four years of MESSENGER magnetometer data. While the analysis presented here is sound and nicely justified, the paper did not convincingly demonstrate that this calibration technique would be

sufficient to perform scientific investigations with the MPO magnetic field observations. The conclusion describes that 780 hours of observations within the magnetosphere are needed to achieve an accuracy better than 1.0 nT; however, many MESSENGER publications including magnetic field data report on signatures that require measurements within this level of uncertainty. Additionally, the manuscript did not describe whether it is expected that MPO will be able to collect compressible fluctuations for 780 hours or more during the mission lifetime. Finally, the major conclusion for application to MPO is that "the 3D mirror mode method developed by (Plaschke et al., 2017) should be applicable to MPO..." but the paper does not describe this method or how it differs from the analysis presented here. The analysis in this paper only provides a single-axis offset – how does this methodology provide vector calibration? Prior to publication, these issues need to be addressed regarding a demonstration of the 3D mirror mode and the ability to use MPO calibrated data with an accuracy of ∼1.0 nT for mission science. Additional comments are listed below:

Paragraph beginning at line 65: The text should also include a description of performing spacecraft rolls as a well-established method for determining offsets. This has been done with routinely with many missions, most recently including MAVEN at Mars and Parker Solar Probe.

Please change all references to MESSENGER into the past tense: Line 100: "MESSENGER is highly" -> "MESSENGER was highly" Line 101: "altitudes ranges" -> "altitudes ranged" Line 101: "form" -> "from" Line 102: "MESSENGER crosses the magnetopause" -> "MESSENGER crossed the magnetopause" Line 107: "MESSENGER is a three-axis-stabilized" -> "MESSENGER was a three-axis-stabilized"

Line 150-152: Please define the mean-field-aligned coordinates system and how it is calculated.

Line 225 – 227: "Note that, although standard deviation of the individual offsets Ozn might be large, a larger number of samples or events helps lower the value of the

standard deviation of the mean offset Ozn (standard error in Table 2)." However, given the small percentage of occurrence rate showed in table 1 – will a large number of samples actually be possible?

Line 270-273: "We find that the offset determination method proposed by Plaschke and Narita (2016) is well applicable to the data from the Hermean environment. It can hence be used for in-flight calibration of the magnetometers onboard Mio and MPO." – While the offset analysis presented here is sound and well-described, it does not demonstrate the application of Plaschke et al (2017) to the MPO dataset, which is most important to derive calibrated vector measurements..

Line 274: "As is been seen in. . ." please revise wording

---

## Referee Comment (RC2) · Anonymous Referee #2 · 14 Apr 2020

Review of "Magnetometer in-flight offset accuracy for the BepiColombo spacecraft" by Schmid, Plaschke, Heyner, Mieth, Anderson, Baumjohann, Volwerk, Matsuoka and Narita

The paper is a bit unusual in that it deals with magnetometer calibrations rather than scientific results. However other than that it is well written and could be published in Annales Geophysicae after suitable corrections. Major Comments The magnetometer offset correction techniques are applied to Alfven waves and mirror mode waves. The general audience will not understand what these waves are and why they would be useful for calibration purposes. Thus I recommend that the authors show examples of each and give the readership some background about their generation mechanisms and properties and why they are in the solar wind and magnetosheath, respectively.

[Figure]

The AG readership should be given some context for why the authors use these two regions of space. The average reader will not know what a mirror mode is. Abstract, lines 11 and 12. A reference should be added for the "mirror mode technique". Line 13. A reference should be given for the "Alfven fluctuation technique". Line 65 and following paragraph. Much of what is written in this paragraph could be deleted without loss. For example it is not necessary to understand the Rosetta null technique involving diamagnetic cavities. However the spin technique is necessary to discuss since you are using it to get two components of your magnetospheric spacecraft magnetometer offsets. Line 74. Give references for the minimum variance method applied to Alfven waves. Line 75. Alfven waves are not always incompressive. See JGRSP, 123, https://doi.org/10.1002/2017JA024203, 2018. This is a misconception in the literature and should be mentioned in this paper. When they are not incompressive, how will the affect your analyses? Please discuss. Line 77. Give references to the mirror mode method here again. The paper is somewhat repetitive. I suggest deleting duplication and shortening the paper considerable. The readership will understand your techniques even if you express them only once.

---

## Author Comment (AC1) · 4 May 2020

First we would like to thank the referee for taking the time to evaluate the manuscript and for the constructive comments which helped to identify potential misunderstandings in the paper.

**Anonymous Referee #1**

The submitted manuscript presents an approach to calibrating offsets on the magnetometers onboard BepiColombo's Mercury Planetary Orbiter and Mercury Magnetospheric Orbiter spacecraft. This calibration analysis includes the use of mirror modewave observations as a method to determine the spin-axis offset. Mio would be able to utilize this approach as a complementary method to an analysis of Alfvenic fluctuations in the pristine solar wind. MPO, on the other hand, will not measure the solarwind and therefore, observations within Mercury's magnetosphere must be used to calibrate measurements. The manuscript presents an analysis of the compressional fluctuations in Mercury's space environment by analyzing four years of MESSENGER magnetometer data. While the analysis presented here is sound and nicely justified, the paper did not convincingly demonstrate that this calibration technique would be sufficient to perform scientific investigations with the MPO magnetic field observations.
We will lower our claims that the method developed and presented in the manuscript is stand-by for immediate applications to the MPO magnetic field data. However, as mentioned in the paper, the method is immediately applicable to the Mio magnetometer because of the need of a single-axis offset determination.

The conclusion describes that 780 hours of observations within the magnetosphere are needed to achieve an accuracy better than 1.0 nT; however, many MESSENGER publications including magnetic field data report on signatures that require measurements within this level of uncertainty.
The magnetic field data, even though not reaching a 1-nT accuracy in the measurements, can certainly be used to a number of publications, but there is no guarantee about the uncertainty or accuracy in the data and care needs to be taken. In particular, if one is interested in finding a magnetic-null (reconnection diffusion region) or low-field phenomena (small-amplitude waves, for example), our method will be of great importance to guarantee how high or low the errors in the data are.

Additionally, the manuscript did not describe whether it is expected that MPO will be able to collect compressible fluctuations for 780 hours or more during the mission lifetime.
Well, a value of 780 hours is a conclusion given by the study; if it is really fulfilled by MPO, we need further studies (MESSENGER data or numerical simulations) and the application of 3D mirror mode method, which is beyond the scope of this paper.

Finally, the major conclusion for application to MPO is that "the 3D mirror mode method developed by (Plaschke et al., 2017) should be applicable to MPO . . ." but the paper does not describe this method or how it differs from the analysis presented here.
The paper primarily focus whether the mirror mode method can be applied in the hermean space environment at all. Indeed the result show that the method can be

directly applied onto Mio. However, the application of the 3-D mirror mode method onto MPO is beyond the scope of the manuscript and is planned to be addressed in a further work. Yet, we will add a paragraph and explain the concept of the 3-D mirror mode method in view of MPO spacecraft.

The analysis in this paper only provides a single-axis offset– how does this methodology provide vector calibration?.
We focus on applications to single-axis offset in the manuscript. Generalization to 3-D offset components is a related yet different issue. We will discuss a paragraph and discuss how to generalize the method to 3 components.

Prior to publication, these issues need to be addressed regarding a demonstration of the 3D mirror mode and the ability to use MPO calibrated data with an accuracy of~1.0 nT for mission science.
There might be a misunderstanding. The manuscript focuses on the calibration method to single-axis offset and discuss the applicability and limits with respect to Mio and MPO magnetometers. We are not claiming that the 1D mirror mode method is readily applicable to MPO magnetometer; Of course, we agree that the 3-D method needs to be developed and tested for MPO; this should be done in a separate paper otherwise the paper has too many goals.

Additional comments are listed below:

Paragraph beginning at line 65: The text should also include a description of performing spacecraft rolls as a well-established method for determining offsets. This has been done with routinely with many missions, most recently including MAVEN at Mars and Parker Solar Probe.
This will be added to the text.

Please change all references to MESSENGER into the past tense: Line 100: "MES-SENGER is highly" -> "MESSENGER was highly" Line 101: "altitudes ranges" -> "altitudes ranged" Line 101: "form" -> "from" Line 102: "MESSENGER crosses the magnetopause" -> "MESSENGER crossed the magnetopause" Line 107: "MESSENGER is a three-axis-stabilized" -> "MESSENGER was a three-axis-stabilized"
This will be changed in the text.

Line 150-152: Please define the mean-field-aligned coordinates system and how it is calculated.
This will be added to the text.

Line 225 – 227: "Note that, although standard deviation of the individual offsets Ozn might be large, a larger number of samples or events helps lower the value of the

standard deviation of the mean offset Ozn (standard error in Table 2)." However, given the small percentage of occurrence rate showed in table 1 – will a large number of samples actually be possible?

As mentioned above, if it is really fulfilled by BepiColombo, we need further information about the final orbit of the spacecraft. At this point, we can only show that for MESSENGER the sample size for the mirror mode method is indeed large enough to reproduce the 1D offset determination which was originally obtained from the Alfvenic fluctuation method.

Line 270-273: "We find that the offset determination method proposed by Plaschke and Narita (2016) is well applicable to the data from the Hermean environment. It can hence be used for in-flight calibration of the magnetometers onboard Mio and MPO."– While the offset analysis presented here is sound and well-described, it does not demonstrate the application of Plaschke et al (2017) to the MPO dataset, which is most important to derive calibrated vector measurements.

Accordingly, we have lowered our claims that the method developed and presented in the manuscript is stand-by for immediate applications to the MPO magnetic field data and thus will delete the sentence.

Line 274: "As is been seen in . . ." please revise wording

This will be changed in the text.

---

## Author Comment (AC2) · 4 May 2020

We like to thank the referee for the helpful comments to improve the quality of the paper and make it more easily comprehensible for the reader.

**Anonymous Referee #2**

The paper is a bit unusual in that it deals with magnetometer calibrations rather than scientific results. However other than that it is well written and could be published in Annales Geophysicae after suitable corrections. Major Comments The magnetometer offset correction techniques are applied to Alfven waves and mirror mode waves. The general audience will not understand what these waves are and why they would be useful for calibration purposes.Thus I recommend that the authors show examples of each and give the readership some background about their generation mechanisms and properties and why they are in the solar wind and magnetosheath, respectively. The AG readership should be given some context for why the authors use these two regions of space. The average reader will not know what a mirror mode is.

The critique is well justified. We will add a paragraph about the Alfven wave and the mirror mode, and introduce how these are used as calibration standard on an elementary level for the benefit to the readers. We add sketches of wave measurements and the use of waves for offset calibration

Abstract, lines 11 and 12. A reference should be added for the "mirror mode technique". Line13. A reference should be given for the "Alfven fluctuation technique".

We think that the Abstract should not contain references since it should be a document on its own.

Line 65 and following paragraph. Much of what is written in this paragraph could be deleted with-out loss. For example it is not necessary to understand the Rosetta null technique involving diamagnetic cavities.

Based on the other referee's suggestions we decided to keep this paragraph, also because it shows that there are various other calibration techniques, which have successfully been applied to previous spacecraft missions (3-axis stabilized and spinning).

However the spin technique is necessary to discuss since you are using it to get two components of your magnetospheric spacecraft magnetometer offsets.

We agree with the referee that a more detail description of the Alfven and mirror mode method is beneficial and thus include additional information about these calibration techniques.

Line 74. Give references for the minimum variance method applied to Alfven waves.

In Belcher (1973) the applied minimum variance method is explained in detail. The reference is given at the end of the sentence.

Line 75. Alfven waves are not always incompressive. See JGRSP,123, https://doi.org/10.1002/2017JA024203, 2018. This is a misconception in the literature and should be mentioned in this paper. When they are not incompressive, how will the affect your analyses? Please discuss.

We fully agree with the referee that Alfven waves are not always incompressive. Only pure Alfvenic fluctuations are strictly incompressible. They are characterized by changes in the magnetic field components while the magnitude of the field stays constant. Particularly in inhomogeneous media such simply classifications are found to be impossible (see Tsurutani (2018) for a review; https://doi.org/10.1002/2017JA024203). In fact, oblique propagation angles to the mean magnetic field, electric field polarization in the plane perpendicular to the mean field, and finite parallel electric field make Alfvenic fluctuations of compressible character by exciting magnetic field fluctuations parallel to the mean field [Narita, 2020; https://doi.org/10.3389/fphy.2020.00166]. These parallel magnetic field fluctuations can be explained within the non-ideal MHD treatment where secondary effects like Hall currents and/or diamagnetic currents are also taken into consideration. These currents flow perpendicular to the mean magnetic field and can cause small-amplitude compressible perturbations. However, in the solar wind the fluctuations of the magnetic field strength (compressible part) are weak compared to the strong fluctuations of the magnetic field vector direction (see e.g. Khabibrakhmanov, 1997; https://doi.org/10.1029/96JA03843). By minimizing the changes of the observed total magnetic field of such fluctuations, it is therefore possible to adjust the magnetometer offsets.

To what extent the incompressibility does affect our analyses is difficult to assess, since for the in-flight offset calibration of MESSENGER additional activities had been performed (e.g. Y-axis spacecraft roll maneuver) which are hard to unravel afterwards. Moreover, to our knowledge the effect on the offset determination has so far never been studied in detail and would be an interesting aspect to look at e.g. for the upcoming BepiColombo in-flight calibration.

Line 77. Give references to the mirrormode method here again.
The reference is given at the beginning of the sentence. Plaschke and Narita (2017)

The paper is somewhat repetitive. I suggest deleting duplication and shortening the paper considerable. The readership will understand your techniques even if you express them only once.
We will go through the paper again and delete sections/sentences, which are repetitive.